# Relationships between sensory processing patterns and metabolic risk factors among community dwelling people with metabolic syndrome: A cross-sectional and correlational research design

Ilada Pomngen[1], Pornpen Sirisatayawong[1], Warunee Kumsaiyai[2], Anuchart Kaunnil[1], Tiam Srikhamjak[1]*

1 Faculty of Associated Medical Sciences, Department of Occupational Therapy, Chiang Mai University, Chiang Mai, Thailand, 2 Faculty of Associated Medical Sciences, Department of Medical Technology, Chiang Mai University, Chiang Mai, Thailand

* tiamsrikhamjak@gmail.com

## Abstract

### Background

Metabolic Syndrome (MetS) increases the risk of other serious health problems, particularly cardiovascular diseases and stroke. Sensory processing patterns (SPPs) are internal factors shaping behaviors and emotions, both healthy and unhealthy. There is a lack of studies directly examining the relationship between the SPPs and metabolic risk factors.

### Method

This study aimed to investigate SPPs and their association with metabolic risk factors in individuals with metabolic syndrome (MetS). One hundred and seventeen individuals with MetS completed questionnaires on demographic characteristics and the Thai Sensory Patterns Assessment-adult version. Data on metabolic risk factors, including fasting blood glucose, blood pressure, and waist circumference, were collected.

### Results

The findings revealed high arousal levels in proprioceptive and auditory senses among the participants. The fasting blood glucose was significantly correlated with a preference in the tactile sense ($r = -0.150$, $P<0.05$), while waist circumference was associated with arousal level in the auditory and smell-taste senses ($r = 0.140$, $-0.160$, $P<0.05$). Moreover, the GLMM revealed that fasting blood glucose was associated with preferences in tactile, vestibular, and proprioceptive senses ($r = -0.481$, $0.726$, $-0.386$, $P<0.05$). Furthermore, diastolic blood pressure was associated with preferences in vestibular sense ($r = 0.099$; $P<0.05$).

**Data Availability Statement:** All relevant data are within the manuscript and its Supporting Information files.

**Funding:** This study was supported by Faculty of Associated Medical Sciences, Chiang Mai University, Thailand, Grant Number AMS-2023.The funders had no role in study design, data collection and analysis, decision to publish, or preparation of the manuscript.

**Competing interests:** The authors have declared that no competing interests exist.

## Conclusion

The SPPs might be related to metabolic risk factors, so it is important to recognize how individual SPPs relate to metabolic risk factors. However, further studies using a larger sample may be needed to deeply explore the mechanisms underlying these associations.

## Introduction

Non-communicable diseases (NCDs) are the leading causes of death worldwide [1]. Currently, one of the main global public-health challenges is metabolic syndrome (MetS), which is defined as a cluster of metabolic abnormalities, such as insulin resistance, visceral obesity, hypertension, and dyslipidemia [2]. Following the occurrence of MetS, people are more likely to be exposed to and develop other serious health problems, in particular cardiovascular diseases [3] and stroke [4,5]. Over the last decades, MetS has been considered a worldwide epidemic because of its increased prevalence in the general population, which is estimated to be at 25% of the world population [6,7] and 20–27% of Thai adults [8]. Understanding contributing factors to minimize the risks of all-cause morbidity and mortality associated with MetS is crucial.

Sensory processing is defined as the ability of the nervous system regarding reception, organization, and response to surrounding sensory input every minute of the day [9]. Each individual is unique and has different ways of processing and responding to sensory information. These ways of processing are called sensory processing patterns (SPPs) [10]. The SPPs reflect how individuals detect, process, and respond in a certain way to environmental and internal sensory stimuli. This includes what sensory stimuli individuals prefer, those they do not prefer, those they have tolerance for, and those they have experienced pleasure with. It also reflects how the amount of sensory stimuli that an individual's brain receives can affect how quickly they notice [11]. In general, people are neurobiologically programmed to adapt their responses to environmental stimuli in appropriate ways. However, some people have more difficulties in processing stimuli, especially those with high or low sensory arousal (or sensory sensitivity) and sensory preference, and they may experience difficulties in adapting their behavior or psychological responses to their environmental stimuli, leading them to avoid or prefer certain sensory experiences [12], which can interfere with performance, participation, and engagement in daily activities, and also impacts health and well-being [13].

SPPs may be linked to the development of MetS due to their influence on human emotions, behaviors, and physiological responses. According to neuroscience, the brain initially serves as a site where sensory information is stored and managed for future use by creating maps of body and environment through sensory experiences and memories accumulated throughout one's lifespan [9,14]. These maps essentially represent one's unique SPPs [11]. The brain needs sensory information to operate and respond, and it relies on these patterns to figure out what to do each day. Our daily decisions, such as what to do, wear, or eat, are related to the amount and types of sensory input we can manage and prefer, as influenced by SPPs [11]. Moreover, when sensory input is processed by the brain it can trigger various physiological responses and hormones are released from the endocrine glands, influencing metabolic processes and potentially contribute to MetS, such as insulin [15] and cortisol [16]. Interestingly, the mismatch between sensory input and SPPs, like sensory overload, can activate stress responses that impact metabolic regulation through the release of cortisol and can further activate the hypothalamic-pituitary-adrenal (HPA) axis and sympathetic nervous system, a condition

previously found in people with Type 1 diabetes [17]. This can lead to changes in heart rate, blood pressure, and energy expenditure, potentially contributing to the development of MetS.

Previous studies found that SPPs are related to health behaviors and psychological factors that contribute to MetS, such as stress [18–20], unhealthy eating behaviors [21], and physical inactivity [22,23]. These associations are mostly found in people with high sensory sensitivity or high sensory arousal patterns. This is because people in this group are more quickly to notice and respond to certain sensory stimuli than those who have typical sensory processing or moderate sensory arousal. Conversely, individuals with low sensory arousal or sensitivity tend to miss and may not respond to certain stimuli. Most of the previous studies have consistently shown that individuals with high sensory sensitivity are more prone to sensory overload in daily life, such as exposure to noise, light, or tactile sensations, leading to chronic stress [19]. Moreover, individuals with greater sensory sensitivity are more likely to engage in emotional eating behaviors triggered by sensory cues from food, such as looking or smelling of food, often as a means of coping with emotions [24]. This occurs because they show higher rates of negative emotion and respond more quickly to food cues than others [18–20]. Additionally, individuals with sensory sensitivity exhibit reduced motivation to engage in physical activity [23], because they easily feel overwhelmed by sensory stimuli associated with exercise environments, such as crowded or noisy gyms or sports events, as well as in environment where intense physical contact is taking place [11].

Based on the evidence provided, the SPPs may indeed play a crucial role in influencing unhealthy psychological and behavioral risks associated with the development and progression of MetS. Previously, although the SPPs have been studied in several groups of people, such as healthy workers [20], healthy adults [18,24], older persons [23], adolescents with Type 1 diabetes [25], and people with multiple sclerosis [26], there currently are no studies specifically focusing on SPPs in people with MetS. Moreover, although previous studies found an association between SPPs and health conditions, such as perceived symptoms of ill health [27] and children's Body Mass Index [28], there are no studies reporting an association between SPPs with metabolic risk variables, such as fasting blood glucose (FBG), blood pressure (BP), waist circumference (WC), or low-density lipoprotein (LDL) and high-density lipoprotein (HDL) cholesterol [2]. Additionally, those studies used assessments, such as the Adolescent/Adult Sensory Profile (AASP) or the Highly Sensitive Person Scale (HSPS), which did not report the SPPs in each specific sensory modality [14,16,19,20,23–26].

The SPPs in adults can be measured through various tools and methods. In past decades, simply asking the person what stimuli they prefer was used [29], but then behavioral observations were later used by presenting specific stimuli and assessing the response of the patient [29,30]. However, behavioral observation requires considerable amounts of time, equipment, and cooperation from the patient. To address these challenges, questionnaires were developed. The most frequently used questionnaire is the Adolescent/Adult Sensory Profile (AASP) [12], which is a self-report used for evaluating behavioral responses to everyday sensory experiences across six sensory domains: taste and smell, movement (vestibular and proprioceptive senses), visual, touch, activity level, and auditory. However, the AASP exhibits some limitations. While its items are organized according to six sensory domains, the scores of all items are summed and interpreted as the SPPs of all sensory domains, including four patterns: low registration, sensation seeking, sensory sensitivity, and sensation avoiding without specifying particular sensory modalities [31]. In fact, individuals may actively seek out some sensory stimuli while avoiding others. Another tool is the Highly Sensitive Person Scale (HSPS), which includes a broad range of items related to high levels of sensory sensitivity or arousal [32]. While interpreting scores of the HSPS can identify individuals with high levels of sensory sensitivity or arousal, it does not provide detailed information about the specific senses involved, which may

not offer specific guidance or interventions tailored to individual needs. Importantly, most of the existing tools have been developed in western countries that might be culturally and contextually inappropriate for other cultures [31].

In Thailand, an assessment tool called the Thai Sensory Patterns Assessment-adult version (TSPA) was developed by modifying the items of AASP and changing the interpreting method by separately interpreting the SPPs in each specific sensory modality: visual, auditory, tactile, smell and taste, proprioceptive, and vestibular senses [33]. This tool not only provides understanding of SPPs and insight into each sensory modality, but also provides better understanding of what specific types of sensory input an individual prefers, and how the arousal of the nervous system operates in detecting and responding to specific sensory stimuli in daily life, which is called sensory preferences and sensory arousal, respectively [33]. Moreover, the TSPA has acceptable and similar levels of internal consistency compared with the ASSP (Cronbach's alpha of TSPA = 0.62–0.89, Cronbach's alpha of ASSP = 0.639–0.775) [12,33]. Understanding sensory preferences and arousal levels via TSPA can indeed be crucial in understanding behavioral responses to specific sensory input in daily living that serves as a specific guide for tailoring interventions to meet individual needs for health promotion or prevention [31]. Although the TSPA has been used in research, such as using TSPA to understand a student's lifestyle related SPPs during the COVID-19 crisis [34] and using TSPA to classify the participants for Mind-Body intervention [35], the TSPA tool has not been used in research to study people with MetS.

To fill the gaps, research that specifically investigates SPPs and their association with metabolic risk factors among individuals with MetS is needed in order to specify what sensory processing patterns in specific sensory modalities are associated with metabolic risk factors. Moreover, in Thailand, the prevalence of MetS is continuously increasing, especially in rural areas, which have a higher prevalence than urban areas [36]. Therefore, the purpose of this study was to examine the SPPs of community-dwelling people with MetS and to determine their association with metabolic risk factors that serve as diagnostic criteria for this condition, such as fasting blood glucose (FBG), systolic blood pressure (SBP) and diastolic blood pressure (DBP), and/or waist circumference (WC). Because the evidence commonly shows that behavioral risks of MetS are sedentary lifestyle [37], physical inactivity [38], and unhealthy eating [39], the hypothesis is that (1) people with MetS have a high sensory arousal in the proprioceptive senses or a high preference in smell and taste senses and that (2) levels of sensory preference and/or sensory arousal in smell-taste and/or proprioceptive senses are expected to be significantly associated with the metabolic risk variables: FBG, SBP and DBP, and/or WC.

## Materials and methods

### Study design

This study used a cross-sectional and correlational design to understand the sensory patterns among people with MetS and examined the association between sensory processing patterns, both sensory preference and sensory arousal, in six sensory modalities (visual, auditory, smell-taste, tactile, vestibular, and proprioceptive senses) and metabolic risk factors, including fasting blood glucose (FBG), systolic blood pressure (SBP), diastolic blood pressure (DBP), and waist circumference (WC).

### Ethics statement

This study has been approved by the Research Ethics Committee of the Faculty of Associated Medical Sciences, Chiang Mai University (No. AMSEC-65EX-071) and was conducted in accordance with the Declaration of Helsinki. Written, informed consent to participate in this study was obtained from all patients and the data was analyzed anonymously.

## Participants

G* power version 3.1 was used to calculate the sample size (p = 0.05, effect size = 0.30, power = 0.90). The minimum total sample size was 112 participants. The test used for the power calculation in G* power was the correlation test, employing a bivariate normal model.

This study collected data among people with MetS who voluntary participated in the annual health checkup and screening for metabolic diseases at Nam Phrae Health Promotion Hospital in Hang Dong District, Chiang-Mai Province, Thailand during January 1st, 2023 to April 30th, 2023. After health checkup and screening, there were 145 people who met the inclusion criteria and were potentially eligible to participate in this study. Of all those eligible participants, 117 participants with MetS (42 males and 75 females), aged between 35–85 years (mean = 54.31 ±10.77), completed the questionnaire.

Inclusion criteria was having three of the metabolic risk factors as defined by harmonized criteria for MetS [2], including elevated FBG ≥100 mg/dL, elevated BP ≥130/85 mmHg, and WC >90 cm for males and >80 cm for females. Exclusion criteria was a history of serious mental illness (depression, schizophrenia, bipolar disorder, or anxiety disorder), diagnosis with sensory impairment or chronic diseases (thyroid, respiratory, liver, kidney, and cardio-vascular diseases), or the presence of cognitive impairments detected by screening. Because this study used self-reported assessments for collecting the data, the screening of cognitive impairment was needed to ensure that participants were able to assess themselves accurately. The Mental State Examination Thai 10 (MSET-10) [40], which was validated and modified from the Mini-Mental State Examination (MMSE)–Thai version was used. This tool has high sensitivity and specificity to detect cognitive impairment, and the items in this tool were modified to be appropriate for Thai culture, particularly for poorly educated people. The total score is 29, and the cutoff score for cognitive impairment was set at 22 for individuals who completed higher than primary school (sensitivity = 100.00% and specificity = 98.40%), 17 for individuals who complete primary school (sensitivity = 100.00% and specificity = 99.30%), or 14 for individuals who are illiterate or cannot read and write (sensitivity = 100.00% and specificity = 94.00%) [41].

## Procedure

Following approval of the study protocol by the Research Ethics Committee of Faculty of Associated Medical Sciences, Chiang Mai University, the participants were drawn from the Nam Phrae Health Promotion Hospital in Hang Dong District, Chiang-Mai Province, Thailand. Purposive sampling was used to recruit the participants. A purposive sampling was the method used to select and recruit specific individuals who are deemed to be the most relevant or most representative of the population being studied by using specific criteria to ensure that participants met the inclusion criteria and were more likely to provide valuable information that aligns with the objectives of the study [42]. The criteria for purposive sampling in this study included: people who (1) have elevated FBG ≥100 mg/dL; (2) have elevated BP ≥130/85 mmHg; (3) have WC >90 cm for males and >80 cm for females; (4) are aged 35 years and above; (5) are able to communicate in the Thai language; (6) are without serious mental illness such as sensory impairment, chronic diseases, or cognitive impairments.

Initially, a physician, who works at Nam Phrae Health Promotion Hospital, screened people who voluntary participated in the annual health checkup and screening for metabolic diseases took place during January 1st, 2023 and April 30th, 2023 by using the purposive sampling criteria. Following the initial screening by a physician, there were a total of 145 individuals with MetS who met the study criteria and were potentially eligible to participate in this study. They were then invited to take part in the study through an advertisement process done by village

health volunteers that involved distributing informational flyers and participant information sheets as well as providing relevant details through word-of-mouth to ensure that individuals had access to all relevant details needed to make a decision about participation in the study. Interested individuals were asked to contact the researcher directly via phone. An assessor asked them to complete the demographic questionnaire and used the MSET-10 to ensure they met the inclusion criteria. Finally, 117 participants (the response rate was 80.69%), who met the criteria and agreed to participate in the study were asked to sign a written informed consent and complete the TSPA. The written informed consent for the study was obtained from all participants, all survey responses were anonymous, and the data were analyzed anonymously.

## Instruments

**Demographic questionnaire.**   This questionnaire was used to gather data, including age, gender, education levels, career, household income, and marital status.

**Measures of metabolic risk factors.**   Data on metabolic risk factors was collected by a trained nurse from the Health Promotion Hospital in order to ensure the quality of the measurements. Fasting blood glucose (FBG) was collected at early morning and after a 12-hour fasting period by fasting capillary blood glucose testing. Blood pressure (BP) was measured in a sitting position after a 10-minute resting period by using an autonomic sphygmomanometer. The mean of the two measures were used to estimate blood pressure [43]. Waist circumference (WC) was measured to the nearest 0.1 cm, midway between the lowest rib and the iliac crest, using a non-elastic circumference measuring tape in a standing position.

**The Thai Sensory Patterns Assessment-adult version (TSPA).**   The TSPA was chosen as the tool for measuring SPPs because the TSPA was designed specifically for Thai adults and can specify sensory processing patterns, including sensory preferences and sensory arousals, in specific sensory modalities [33], while other tools, such as the AASP and HSPS, cannot provide detailed information about SPPs in a specific sense.

The Thai Sensory Patterns Assessment (TSPA) tool was developed in Thailand by adapting the items of Dunn's sensory profile into two parts: sensory preferences and sensory arousals. A sensory preference is defined as the type of sensory stimulus that tends to make individuals feel affirmed and more comfortable, and is even pleasurable when receiving sensory input. Sensory arousals are defined as the level of the nervous system's alertness to detect, register, and respond to sensory input in daily life.

The TSPA is a self-report measurement, which consists of 60 items and is divided into two parts: one for sensory preferences with 35 items and the second for sensory arousals with 25 items. Each part is divided into six categories based on the type of sensory modalities: visual, auditory, smell and taste, tactile, proprioceptive, and vestibular senses, as confirmed by confirmatory factor analysis (CFA) [33]. In this self-reported assessment, the participants indicated by answering questions on each item about how often they respond to the sensory event in everyday life by rating on a 5-Likert scale in each item: 1 = never, 2 = seldom, 3 = occasionally, 4 = frequently, 5 = always in the part one: sensory preferences. In part two the rating scale for sensory arousals was as follows: 1 = never, 2 = seldom, 3 = occasionally, 4 = frequently, 5 = always in items with high arousal, and 5 = never, 4 = seldom, 3 = occasionally, 2 = frequently, 1 = always in items with low arousal.

The examples of items include: smell-taste sensory preference ("Like eating sweets or smelling food"); proprioceptive sensory preference ("Prefer doing activities that involve pulling, pushing, banging, such as boxing, jumping rope, lifting weights, etc."); smell-taste sensory arousal ("Can eat food that has a strong or pungent smell"); proprioceptive sensory arousal

("Often break or damage items because of excessive force while handling"). The scores for each sensory modality were summed, interpreted, and reported separately as a percentage (below 25% = low, at 25–75% = moderate, and above 75% = high sensory preferences or levels of sensory arousals).

For psychometric properties, the content validity was determined by examination of the index of item–objective congruence (IOC) of Part One: sensory preferences and Part Two: sensory arousals ranging from 0.60 to 1.00. The internal consistency reliability was an α coefficient of 0.89 for Part One and 0.62 for Part Two. The test–retest reliability with intraclass correlation coefficient (ICC) method was 0.93 in Part One and 0.77 in Part Two. The construct validity by confirmatory factor analysis (CFA) found that there were six factors in each part, and all of the items had a high factor loading (ranging from 0.422 to 0.815 for Part One and 0.484 to 0.849 for Part Two). In conclusion, the TSPA was both valid and reliable at an acceptable level [33].

To avoid bias from self-reporting, before participants completed the questionnaire, researchers provided them with instructions regarding the study's purpose, the importance of honesty and accuracy in responses, and the anonymity of their answers. To avoid missing data, after participants completed the questionnaire, the researcher rechecked the responses to make sure that all of items were completed.

## Data analysis

Statistical analyses were undertaken using the Statistical Package for Social Science (SPSS) software for Windows-Version 20 (SPAA inc., USA). Descriptive statistics were used to analyze data on participant demographical characteristics, metabolic risk factors, and sensory processing patterns. The results were presented as means and standard deviations (SD) for continuous variables (sensory processing pattern and metabolic risk variables), and as percentages for categorical variables (sex, education levels, career, household income, marital status, and age groups). The Kolmogorov-Smirnov test was applied to determine the normal distribution of variables. The test found that sensory patterns and metabolic risk factors were not distributed normally. Differences between age groups and sex for metabolic risk factors and MSET-10 scores were tested using Kruskal-Wallis Test and Mann-Whitney U Test, respectively. Kendall's Tau was used to examine the relationship between sensory processing patterns, including sensory preferences and arousals in specific modalities, and metabolic risks factors including SBP and DBP, WC, and FBG. The Generalized Linear Mixed Model (GLMM) was employed to examine the association between each metabolic risk factor and sensory patterns in specific sensory modalities. The sensory patterns, both sensory preferences and sensory arousals, in each specific sensory modality were used as explanatory variables, which were included in a single model. Each metabolic risk factor, including SBP, DBP, WC and FBG, was used as target variables (dependent variables) in a separate analysis. SPSS software (Version 20) was used to conduct the GLMM. For the GLMM, a Gaussian family distribution was chosen based on the continuous nature of the dependent variables (metabolic risk variables). The identity link function was selected for each regression model to effectively model the linear relationship between predictors (sensory processing patterns) and outcome variables (metabolic risk factors). Model fit was assessed using Akaike Information Criterion (AIC) and Bayesian Information Criterion (BIC), both based on -2 Log Likelihood values, to compare the adequacy of different models. Statistical significance was determined with p-values below 0.05.

## Results

After screening, there were a total of 145 individuals with MetS who met the study criteria and were potentially eligible to participate in this study. After an advertisement process, a total of

117 participants agreed to participate in the study and were included in the analysis, while 28 people decided not to participate in the study because of individual reasons regarding time limitations and their personal interests. There was no missing data and there were no participants with incomplete data.

## Demographical characteristics

The demographic characteristics of 117 participants with MetS are presented in Table 1. The majority of participants were female (64.1%) aged between 35 and 64 years old (61.5%). In addition, the findings reported that most participants had undertaken primary school (50.3%), had low household income of 5,001–10,000 THB per month (29.9%), and more than 50 percent of all the participants were married (65.8%).

The metabolic risk factors and the MSET-10 scores of 117 participants with MetS are presented in Table 2. All participants were without cognitive impairment (mean = 26.98±1.59). The majority of participants were prediabetes (76.07%), stage 1 of hypertension (SBP = 80.34%, DBP = 69.24%), and wrist circumference was 90–95 cm. in males and 80–85 cm. in females (52.14). In addition, there were no significant differences between sex and age in the metabolic risk factors and MSET-10 scores, except for WC that found significant difference between sex (p = <0.001).

**Table 1. Demographic characteristics in people with MetS (n = 117).**

| Variable | All (n = 117) n (%) | Male (n = 42) n (%) | Female (n = 75) n (%) |
|---|---|---|---|
| Sex | | | |
| Male | 42 (35.9) | 42 (35.9) | 75 (64.1) |
| Female | 75 (64.1) | | |
| Education Levels | | | |
| Under primary school | 17 (14.5) | 6 (14.3) | 11 (14.7) |
| Primary school | 60 (51.3) | 15 (35.7) | 45 (60.0) |
| High school | 32 (27.4) | 16 (38.1) | 16 (21.3) |
| Bachelor's degree | 8 (6.8) | 5 (11.9) | 3 (4.0) |
| Career | | | |
| No career | 22 (18.8) | 6 (14.3) | 16 (21.3) |
| Contractors | 45 (38.5) | 19 (45.2) | 26 (34.7) |
| Merchant | 25 (21.4) | 5 (11.9) | 20 (26.7) |
| Government official | 4 (3.4) | 2 (4.8) | 2 (2.7) |
| Agriculture | 12 (10.3) | 7 (16.7) | 5 (6.7) |
| Officer | 8 (6.8) | 3 (7.1) | 5 (6.7) |
| Others: Owner | 1 (0.9) | 0 (0) | 1 (1.3) |
| Household income (THB per month) | | | |
| Less than 5,000 | 19 (16.2) | 8 (19.0) | 11 (14.7) |
| 5,001–10,000 | 35 (29.9) | 10 (23.8) | 25 (33.3) |
| 10,001–15,000 | 27 (23.1) | 9 (21.4) | 18 (24.0) |
| 15,001–20,000 | 14 (12.0) | 6 (14.3) | 8 (10.7) |
| 20,001–25,000 | 11 (9.4) | 3 (7.1) | 8 (10.7) |
| More than 25,000 | 11 (9.4) | 6 (14.3) | 5 (6.7) |
| Marital status | | | |
| Single | 18 (15.4) | 9 (21.4) | 9 (12.0) |
| Married | 77 (65.8) | 28 (66.7) | 49 (65.3) |
| Divorced | 9 (7.7) | 2 (4.8) | 7 (9.3) |
| Widowed | 13 (11.1) | 3 (11.9) | 10 (13.3) |
| Age | | | |
| 35–44 years | 19 (16.2) | 9 (21.4) | 10 (13.3) |
| 45–64 years | 72 (61.5) | 25 (59.5) | 47 (62.7) |
| 65 years and over | 26 (22.2) 54.31±10.77 | 8 (7.1) | 18 (24.0) |
| Age (Mean ± SD) | | 53.83±11.74 | 54.57±10.26 |

**Table 2. Metabolic risk factors and MSET-10 scores in people with MetS (n = 117).**

| Variable | All (n = 117) | Male (n = 42) | Female (n = 75) | P value Between sex | P value Between age |
|---|---|---|---|---|---|
| MSET-10 (Mean ± SD) | 26.98±1.59 | 27.04±1.41 | 26.94±1.69 | 0.840[a] | 0.211[b] |
| FBG | 115.61±18.17 | 114.14±17.04 | 116.44±18.83 | 0.721[a] | 0.515[b] |
| Prediabetes (100–125 mg/dL) | 107.11±8.73 | 106.15±7.66 | 107.64±9.29 | 0.741[a] | 0.297[b] |
| Mean ± SD | 89 (76.07) | 32 (76.19) | 57 (76.00) | 0.302[a] | 0.017[b*] |
| n (%) | 142.64±13.20 | 139.70±13.00 | 144.27±13.39 | | |
| Diabetes (≥ 126 mg/dL) | 28 (23.93) | 10 (23.81) | 18 (24.00) | | |
| Mean ± SD | | | | | |
| n (%) | | | | | |
| SBP | 135.06±6.56 | 134.76±6.10 | 135.22±6.83 | 0.705[a] | 0.057[b] |
| Stage 1 (130–139 mmHg) | 132.41±2.90 | 132.32±2.87 | 132.46±2.94 | 0.688[a] | 0.409[b] |
| Mean ± SD | 94 (80.34) | 34 (80.95) | 60 (80.00) | 0.922[a] | 0.608[b] |
| n (%) | 145.86±6.24 | 145.12±5.22 | 146.26±6.87 | | |
| Stage 2 (≥ 140 mmHg) | 23 (19.66) | 8 (19.05) | 15 (20.00) | | |
| Mean ± SD | | | | | |
| n (%) | | | | | |
| DBP | 86.88±3.30 | 86.69±2.89 | 86.98±3.51 | 0.954[a] | 0.230[b] |
| Stage1 (80–89 mmHg) | 85.09±1.85 | 85.10±1.91 | 85.09±1.83 | 0.995[a] | 0.167[b] |
| Mean ± SD | 81 (69.24) | 29 (69.05) | 52 (69.33) | 0.489[a] | 0.685[b] |
| n (%) | 90.88±2.06 | 90.23±0.59 | 91.26±2.49 | | |
| Stage 2 (≥ 90 mmHg) | 36 (30.76) | 13 (30.95) | 23 (30.67) | | |
| Mean ± SD | | | | | |
| n (%) | | | | | |
| WC | 91.36 ± 9.37 | 94.09 ± 6.44 | 89.82 ± 10.38 | <0.001[a**] | 0.308[b] |
| 90–95 cm (for male) or 80–85 cm (for female) | 86.01±4.95 | 90.63±2.07 | 81.54±1.72 | <0.001[a**] | 0.658[b] |
| Mean ± SD | 61 (52.14) | 30 (71.43) | 31 (41.33) | 0.001[a**] | 0.206[b] |
| n (%) | 97.17±9.60 | 102.75±5.39 | 95.65±9.97 | | |
| >95 cm (for male) or > 85 cm (for female) | 56 (47.86) | 12 (28.57) | 44 (58.67) | | |
| Mean ± SD | | | | | |
| n (%) | | | | | |

FBG, Fasting blood glucose; SBP, Systolic blood pressure; DBP, Diastolic blood pressure; WC, Wrist circumference.

[a]p-values were calculated using the Mann-Whitney U Test; p ≤ 0.05*, P ≤ 0.01**, p ≤ 0.001***.

[b]p-values were calculated using Kruskal-Wallis Test, p ≤ 0.05*, p ≤ 0.01**, p ≤ 0.001***.

## Sensory processing patterns

The scores of sensory processing patterns from the TSPA, including sensory preference and sensory arousals in specific sensory modalities, are presented in Table 3. Most of the participants had moderate preferences (median 30.00–60.00) and moderate arousals (median 40.00–73.00) in all senses. There were high arousal levels (median 80.00 and 82.85), only in proprioceptive and auditory senses. In addition, the highest score of sensory preferences was found in the auditory sense (median 64.00), while the lowest was found in the vestibular sense (median 30.00).

**Correlation between sensory patterns and metabolic risk factors.** The correlation between sensory processing patterns and metabolic risk factors is presented in Table 4. The metabolic risks, in particular fasting blood glucose (FBG), were correlated negatively with tactile preferences (r = -0.150, P<0.05), while there was no significant correlation between other metabolic risks (SBP, DBP, and WC) and sensory preferences in any sensory modalities. Moreover, only WC was correlated positively with arousal levels in the auditory sense and negatively with arousal levels in the smell-taste senses (r = 0.140, -0.160, P<0.05), while there was no significant correlation between other metabolic risks (FPG, SBP, and DBP) and sensory arousals in any sensory modality.

**Table 3. Scores of sensory processing patterns from the TSPA in people with MetS (n = 117).**

| Sensory patterns | Minimum | Maximum | Median | Mean | Std. Deviation | Levels of sensory preferences/ sensory arousals |
|---|---|---|---|---|---|---|
| *Sensory preferences* | | | | | | |
| Visual | 20.00 | 95.00 | 60.00 | 57.38 | 16.50 | Moderate |
| Auditory | 20.00 | 100.00 | 64.00 | 67.00 | 17.17 | Moderate |
| Smell and taste | 28.86 | 88.80 | 59.94 | 57.86 | 11.90 | Moderate |
| Tactile | 20.00 | 100.00 | 40.00 | 42.82 | 15.52 | Moderate |
| Vestibular | 20.00 | 60.00 | 30.00 | 33.12 | 11.03 | Moderate |
| Proprioceptive | 20.00 | 80.00 | 36.00 | 38.11 | 13.06 | Moderate |
| *Sensory arousals* | | | | | | |
| Visual | 20.00 | 100.00 | 73.33 | 70.86 | 21.06 | Moderate |
| Auditory | 20.00 | 100.00 | 80.00 | 78.15 | 13.00 | High |
| Smell and taste | 20.00 | 100.00 | 60.00 | 61.53 | 18.45 | Moderate |
| Tactile | 20.00 | 100.00 | 40.00 | 41.98 | 16.36 | Moderate |
| Vestibular | 20.00 | 100.00 | 40.00 | 39.29 | 17.87 | Moderate |
| Proprioceptive | 40.00 | 100.00 | 82.85 | 81.32 | 14.29 | High |

Levels of sensory preferences/sensory arousals interpreted by TSPA scores, Low = a percentage of score below 25%, Moderate = a percentage of score at 25–75%, High = a percentage of score above 75%.

**Table 4. Correlation between sensory processing patterns and metabolic risk factors (n = 117).**

| Metabolic risk factors | Sensory preferences | | | | | |
|---|---|---|---|---|---|---|
| | Visual | Auditory | Smell and taste | Tactile | Vestibular | Proprioceptive |
| FBG | | | | | | |
| Correlation Coefficient | 0.045 | 0.108 | -0.020 | -0.150* | 0.075 | -0.026 |
| Sig (2-tailed) | 0.510 | 0.109 | 0.761 | 0.027 | 0.268 | 0.695 |
| SBP | | | | | | |
| Correlation Coefficient | -0.026 | -0.064 | -0.089 | -0.084 | -0.118 | -0.096 |
| Sig (2-tailed) | 0.708 | 0.356 | 0.195 | 0.231 | 0.089 | 0.171 |
| DBP | | | | | | |
| Correlation Coefficient | -0.136 | 0.002 | -0.053 | -0.052 | 0.129 | 0.060 |
| Sig (2-tailed) | 0.061 | 0.974 | 0.459 | 0.478 | 0.075 | 0.411 |
| WC | | | | | | |
| Correlation Coefficient | 0.051 | 0.069 | 0.091 | 0.040 | 0.069 | 0.105 |
| Sig (2-tailed) | 0.488 | 0.302 | 0.166 | 0.550 | 0.302 | 0.118 |
| Metabolic risk factors | Sensory arousals | | | | | |
| | Visual | Auditory | Smell and taste | Tactile | Vestibular | Proprioceptive |
| FBG | | | | | | |
| Correlation Coefficient | -0.021 | -0.095 | 0.048 | 0.049 | 0.050 | -0.034 |
| Sig (2-tailed) | 0.760 | 0.155 | 0.482 | 0.474 | 0.469 | 0.615 |
| SBP | | | | | | |
| Correlation Coefficient | 0.036 | 0.056 | 0.036 | -0.015 | -0.020 | -0.011 |
| Sig (2-tailed) | 0.607 | 0.422 | 0.612 | 0.835 | 0.776 | 0.876 |
| DBP | | | | | | |
| Correlation Coefficient | -0.095 | -0.068 | 0.024 | -0.001 | 0.034 | -0.051 |
| Sig (2-tailed) | 0.192 | 0.345 | 0.747 | 0.986 | 0.646 | 0.482 |
| WC | | | | | | |
| Correlation Coefficient | 0.071 | 0.140* | -0.160* | -0.109 | -0.039 | 0.078 |
| Sig (2-tailed) | 0.291 | 0.036 | 0.017 | 0.106 | 0.572 | 0.238 |

FBG, Fasting blood glucose; SBP, Systolic blood pressure; DBP, Diastolic blood pressure; WC, Wrist circumference.

p-values were calculated using Kendall's Tau; $p \leq 0.05^*$, $p \leq 0.01^{**}$, $p \leq 0.001^{***}$.

As presented in Table 5, the general linear mixed model revealed that sensory processing patterns were the only sensory preferences associated with metabolic risk factors, particular FBG and DPB. Specifically, FBP was negatively associated with sensory preferences in the

**Table 5. Relationship between sensory processing patterns and metabolic risk factors (n = 117).**

| | source | coefficient | SE | t | F | Sig. | 95% confidence interval | |
|---|---|---|---|---|---|---|---|---|
| | | | | | | | lower | upper |
| FBG | Intercept | 100.588 | 22.702 | 4.431 | 1.996 | <0.001*** | 55.569 | 145.608 |
| | P-visual | 0.063 | 0.126 | 0.496 | 0.246 | 0.621 | -0.188 | 0.313 |
| | P-auditory | 0.181 | 0.118 | 1.532 | 2.347 | 0.129 | -0.053 | 0.415 |
| | P-smell and taste | 0.094 | 0.194 | 0.484 | 0.234 | 0.629 | -0.291 | 0.478 |
| | P-tactile | -0.481 | 0.140 | -3.429 | 11.755 | 0.001** | -0.760 | -0.203 |
| | P-vestibular | 0.726 | 0.228 | 3.187 | 10.154 | 0.002** | 0.274 | 1.178 |
| | P-proprioceptive | -0.386 | 0.184 | -2.105 | 4.431 | 0.038* | -0.750 | -0.022 |
| | A-visual | -0.030 | 0.088 | -0.338 | 0.114 | 0.736 | -0.204 | 0.145 |
| | A-auditory | -0.262 | 0.154 | -1.704 | 2.904 | 0.091 | -0.568 | 0.043 |
| | A-smell and taste | -0.165 | 0.109 | 1.520 | 2.310 | 0.132 | -0.050 | 0.381 |
| | A-tactile | 0.112 | 0.114 | 0.988 | 0.975 | 0.326 | -0.113 | 0.337 |
| | A-vestibular | 0.046 | 0.108 | 0.427 | 0.182 | 0.670 | -0.167 | 0.259 |
| | A-proprioceptive | 0.136 | 0.145 | 0.940 | 0.883 | 0.349 | 0.151 | 0.424 |
| SBP | Intercept | 138.343 | 8.762 | 15.790 | 0.657 | <0.001*** | 120.968 | 155.718 |
| | P-visual | 0.038 | 0.049 | 0.779 | 0.657 | 0.438 | -0.059 | 0.135 |
| | P-auditory | -0.011 | 0.046 | -0.239 | 0.057 | 0.812 | -0.101 | 0.080 |
| | P-smell and taste | -0.082 | 0.075 | -1.097 | 1.204 | 0.275 | -0.231 | 0.066 |
| | P-tactile | 0.011 | 0.054 | 0.197 | 0.039 | 0.845 | -0.097 | 0.118 |
| | P-vestibular | -0.084 | 0.088 | -0.960 | 0.922 | 0.339 | -0.259 | 0.090 |
| | P-proprioceptive | -0.028 | 0.071 | -0.391 | 0.153 | 0.697 | -0.168 | 0.113 |
| | A-visual | 0.016 | 0.034 | 0.486 | 0.236 | 0.628 | -0.051 | 0.084 |
| | A-auditory | 0.031 | 0.059 | 0.515 | 0.266 | 0.607 | -0.087 | 0.148 |
| | A-smell and taste | -0.005 | 0.042 | -0.123 | 0.015 | 0.902 | -0.088 | 0.078 |
| | A-tactile | 0.030 | 0.044 | 0.679 | 0.462 | 0.498 | -0.057 | 0.117 |
| | A-vestibular | -0.033 | 0.041 | -0.801 | 0.642 | 0.425 | -0.116 | 0.049 |
| | A-proprioceptive | 0.003 | 0.056 | 0.050 | 0.003 | 0.960 | -0.108 | 0.114 |
| DBP | Intercept | 90.657 | 4.223 | 21.469 | 1.474 | <0.001*** | 82.283 | 99.031 |
| | P-visual | -0.036 | 0.023 | -1.535 | 2.357 | 0.128 | -0.083 | 0.011 |
| | P-auditory | -0.004 | 0.022 | -0.194 | 0.038 | 0.846 | -0.048 | 0.039 |
| | P-smell and taste | -0.013 | 0.036 | -0.354 | 0.125 | 0.724 | -0.084 | 0.059 |
| | P-tactile | -0.030 | 0.026 | -1.165 | 1.358 | 0.247 | -0.082 | 0.021 |
| | P-vestibular | 0.099 | 0.042 | 2.339 | 5.472 | 0.021* | 0.015 | 0.183 |
| | P-proprioceptive | -0.035 | 0.034 | -1.003 | 1.067 | 0.304 | -0.103 | 0.032 |
| | A-visual | -0.012 | 0.016 | -0.715 | 0.511 | 0.476 | -0.044 | 0.021 |
| | A-auditory | -0.050 | 0.029 | -1.743 | 3.037 | 0.084 | -0.107 | 0.007 |
| | A-smell and taste | 0.009 | 0.020 | 0.438 | 0.192 | 0.663 | -0.031 | 0.049 |
| | A-tactile | 0.001 | 0.021 | 0.027 | 0.001 | 0.979 | -0.041 | 0.042 |
| | A-vestibular | 0.012 | 0.020 | 0.576 | 0.331 | 0.566 | -0.028 | 0.051 |
| | A-proprioceptive | 0.029 | 0.027 | 1.087 | 1.182 | 0.279 | -0.024 | 0.083 |
| WC | Intercept | 79.130 | 12.346 | 6.409 | 0.916 | <0.001*** | 54.647 | 103.613 |
| | P-visual | 0.017 | 0.069 | 0.242 | 0.058 | 0.809 | -0.119 | 0.153 |
| | P-auditory | 0.017 | 0.064 | 0.270 | 0.073 | 0.787 | -0.110 | 0.145 |
| | P-smell and taste | 0.078 | 0.105 | 0.742 | 0.550 | 0.460 | -0.131 | 0.287 |
| | P-tactile | -0.016 | 0.076 | -0.205 | 0.042 | 0.838 | -0.167 | 0.136 |
| | P-vestibular | -0.067 | 0.124 | -0.542 | 0.294 | 0.589 | -0.313 | 0.178 |
| | P-proprioceptive | 0.159 | 0.100 | 1.594 | 2.540 | 0.114 | -0.039 | 0.357 |
| | A-visual | -0.011 | 0.048 | -0.239 | 0.057 | 0.812 | -0.106 | 0.083 |
| | A-auditory | 0.137 | 0.084 | 1.638 | 2.684 | 0.104 | -0.029 | 0.303 |
| | A-smell and taste | -0.056 | 0.059 | -0.943 | 0.889 | 0.348 | -0.173 | 0.062 |
| | A-tactile | -0.065 | 0.062 | -1.051 | 1.104 | 0.296 | -0.187 | 0.058 |
| | A-vestibular | -0.013 | 0.058 | -0.223 | 0.050 | 0.824 | -0.129 | 0.103 |
| | A-proprioceptive | -0.010 | 0.079 | -0.128 | 0.016 | 0.898 | -0.167 | 0.146 |

FBG, Fasting blood glucose; SBP, Systolic blood pressure; DBP, Diastolic blood pressure; WC, Wrist circumference.

p-value were calculated using Generalized Linear Mixed Model; p ≤ 0.05*, p ≤ 0.01**, p ≤ 0.001***.

tactile and proprioceptive senses (R = -0.481, -0.386; p = 0.001, 0.038, respectively), and positively associated with sensory preference in the vestibular sense (R = 0.726; p = 0.002). Moreover, DBP was positively associated with sensory preference in the vestibular sense (R = 0.099; p = 0.021). However, as shown in Table 5, SBP and WC were not significantly associated with either sensory preferences or sensory arousals in any sensory modality.

## Discussion

This study aimed to investigate the sensory processing patterns (SPPs) among people with MetS in the community and examined the association between the SPPs and the metabolic risk factors, including fasting blood glucose (FBG), systolic blood pressure (SBP) and diastolic blood pressure (DBP), and/or waist circumference (WC). Key findings revealed that high arousal levels in the proprioceptive and auditory senses were found among the participants, and that the SPPs in tactile, auditory, smell-taste, vestibular and proprioceptive senses were significantly associated with metabolic risk factors, particularly FBG, WC, and DBP.

### The sensory processing patterns of people with MetS

This study was the first investigation of the SPPs among people with MetS in this community. The results from this study supported our hypothesis that a pattern of high sensory arousal in proprioceptive sense was found among the participants with MetS. However, a pattern of high preference in smell and taste sense was not found among the participants, while they showed moderate sensory preference in the smell-taste sense.

The findings were consistent with a previous study which showed that individuals with high sensitivity to sensory stimuli were also found to have metabolic health problems, such as Type 1 diabetes [25]. However, the previous study did not look into sensory processing patterns in specific sensory modalities [25], and no research on people with MetS has been conducted. Thus, these findings added to the earlier understanding by specifying sensory processing patterns in specific sensory modalities among people with MetS by using the TSPA tool, which discovered that high levels of arousal in the proprioceptive and auditory senses were found in participants with MetS.

Moreover, the primary characteristic of people with high levels of arousal in the proprioceptive sense is that they tend to more quickly detect certain proprioceptive stimuli compared to others [11]. Consequently, they experience certain physical activities as overwhelming or uncomfortable due to the intense sensory input involved in forceful movements, leading to negative experiences, such as discomfort, pain, or injury, thus allowing them to avoid engaging in physical activities. This was consistent with the findings of previous studies that showed that the majority of people with MetS represent a sedentary activity type and that there is a decrease in physical activity [44]. Similarly, previous research indicated a link between sensory sensitivity pattern in the AASP and reduced physical activity among healthy older adults, but this research doesn't specify which sensory modality has the most significant impact [23].

Therefore, providing specific information on sensory processing patterns in specific senses can greatly assist therapists in understanding individual behaviors better and helps to tailor targeted intervention aimed at promoting healthier routines, habits, and daily environments that minimize the risks of MetS. This can be applied by promoting increased physical activities in ways that match their sensory patterns. For instance, low-intensity proprioceptive activities like yoga, walking, or muscle stretching, could be appropriate for individuals with high sensory arousal.

## Association between sensory processing patterns and metabolic risk factors

The results from this study supported our hypothesis that levels of sensory preference and/or sensory arousal in smell-taste and/or proprioceptive senses were significantly associated with the metabolic risk variables. Although the correlations were low, the findings suggest potential associations that might enhance awareness among healthcare professionals regarding the relationship between sensory processing patterns and metabolic health, which might help in the development of more comprehensive treatment plans or further research in this area.

The GLMM revealed the association between fasting blood glucose (FBG) levels and sensory preference in the proprioceptive sense. This suggests that individuals with higher FBG levels tend to exhibit lower sensory preference for proprioceptive senses. The characteristic of people with low sensory preference in proprioceptive sense is that they tend to not prefer or decline to do any activity requiring high-intensity of proprioceptive senses, such as certain sports, weight training, or physical activities involving joint compression or forceful movement. Similarly, prior research has indicated that participants with MetS showed higher amounts of sedentary time compared to those without MetS [45], and more time spent in sedentary behavior was predictive of significant increases in hyperglycemic time [46]. Therefore, the findings emphasize the need for the development of individualized interventions aimed at reducing sedentary time to decrease hyperglycemia.

Specifically, in wrist circumference (WC), the significantly negative relationship between WC and arousal levels in smell-taste senses were both consistent with and different from previous studies. Naish and Harris [47], found significantly higher food intake, particularly chocolate, in individuals with high sensory arousal compared to those with low arousal. This previous finding suggests that people with high sensory arousal are more likely to exhibit increased WC which contrasts with the findings of this study. However, this previous study faced limitations in specifying which senses predominantly influence food intake, because factors beyond taste preference, such as visual appraisal, smell, and texture, also can influence eating behaviors. Nevertheless, the findings of this study align with other studies indicating that individuals with low arousal level in smell-taste senses prefer intensely flavored or condiment-rich foods, such as those that are sweet or salty [48–51]. This, in turn, contributes to weight gain and increased WC [52]. Based on the findings, healthcare professionals should consider individuals with low arousal levels in the smell-taste sense when offering dietary advice or interventions. Understanding how sensory processing patterns influence food choices, allows for tailored recommendations to support individuals in developing sensory-based strategies for healthier food choices in their daily lives. The incorporation of mindful eating practices may prove beneficial for this group in preventing increased WC.

Previous research has suggested a link between high sensory arousal and obesity indicators, such as BMI among children [28]. However, our findings revealed a low positive association between waist circumference (WC) and sensory arousal specifically in the auditory sense. Due to the nature of our data and the observed low correlation, we cannot draw a direct link between WC and sensory arousal in the auditory sense. Consequently, further research is necessary to confirm and establish a causal relationship.

Furthermore, the findings revealed a negative relationship between FBG levels and sensory preference in the tactile senses. This finding suggests that higher levels of FBG are associated with a lower preference for tactile sense, while a previous study found that tactile sensitivity was significantly associated with an increase in BMI among children [28]. As a result, sensory processing in the tactile sense might potentially be involved in metabolic health not only in BMI but also in FBG. The authors of previous studies stated that sensory processing in the tactile sense may be an important contributor to food acceptance because sensory properties of

foods include not only taste and olfactory characteristics, but also visual and tactile ones [28]. Because sensory sensitivity is linked to perception and preference for food texture [53], people with low preference for tactile sensations tend to dislike specific textures (e.g., mushy, crunchy, slimy). This often leads to the avoidance of specific kinds of foods that possess these textures [54]. For example, some individuals dislike mushy textures, leading them to avoid foods like bananas or cooked vegetables. In this regard, eating problems in adults, such as picky eating, selective eating, or inflexible eating behaviors, have been linked to less healthy food choices [55], and an unbalanced diet can have long-term effects on metabolic outcomes related to FBG levels. Additionally, sensory stimuli that do not match a person's preferences can be overwhelming [11], especially tactile senses, which are strongly linked to emotion and security [56]. People with a low tactile preference may be easily overwhelmed by certain sensory environments, such as crowded areas, food textures, and clothing materials. This sensory overload can activate the sympathetic-adrenal-medullary (SAM) and hypothalamic-pituitary-adrenal (HPA) axes, resulting in elevated cortisol levels. Over time, this chronic activation can contribute to metabolic abnormalities [57,58], such as insulin resistance, which causes high FBG levels. The findings suggest that increasing awareness of the potential impact of the tactile senses on metabolic health, as well as implementing sensory-based strategies for dealing with the unpleasantness of sensory events in daily life, are critical.

Finally, the study found positive associations between sensory preferences in vestibular sense and metabolic risk factors, including FBG and DBP. Previous studies indicated the role of the vestibular system connected to the vestibulosympathetic reflexes in regulating blood pressure [59]. The effect of vestibular stimulation varies depending on its type and intensity. Previous research has shown that gentle vestibular stimulation, such as a gentle front and back swinging, can reduce blood pressure and blood glucose levels [60,61]. Conversely, sudden movements or changes in posture can trigger responses in the sympathetic nervous system, resulting in an increase in blood pressure and blood glucose [62]. Individuals with a high sensory preference in the vestibular sense typically exhibit characteristics such as a tendency to seek out experiences that provide vestibular stimulation, whereas the TSPA items involved both gentle and sudden changes in posture. As a result, further research on this association is needed to establish a clearer link, particularly specifying sensory preferences in specific types of vestibular stimulation associated with FBP and/or DBP.

In summary, this is the first study to discover that sensory processing patterns are associated with metabolic risk factors, such as FBG, DBP, and WC. Although, these correlations were low, these findings suggest the possible role of sensory processing patterns in metabolic risk factors. However, additional research, preferably with longitudinal data, is crucial to achieve a clearer understanding of the causal links between sensory processing patterns and metabolic risk factors. Importantly, knowledge regarding sensory processing patterns of people with MetS may help healthcare professionals increase awareness about the potential impact of sensory processing patterns on metabolic health. Moreover, by considering an individual's sensory processing patterns, healthcare professionals can develop personalized healthcare strategies based on individual sensory needs necessary for designing environments or they can provide dietary advice and healthy lifestyle recommendations that align with individual sensory processing patterns, aimed to mitigate specific MetS risk factors.

## Study limitations

However, this study had some limitations when applying these findings in practice. Although the significant associations between sensory processing patterns and metabolic risk factors were found, these correlations are still low. Moreover, sensory processing patterns are complex

because each person differs in terms of their needs and their ability to not being overwhelmed while responding to sensory input. Therefore, to effectively implement the findings of this study into practice, it is suggested that knowledge gained should be combined with other relevant information, such as observations of how patients respond to sensory input in their daily lives or through an in-depth interview, in order to provide deep and reliable information. In addition, this study has some methodological limitations, such as a limited generalizability due to a relatively small sample size through the use of a purposive sampling method, and reliance on self-reported data from the TSPA tool, which can be subject to bias or inaccuracies that might impact the study's reliability. To further validate the findings and enhance the study's generalizability, a larger-scale study with more diverse participants is necessary. This would provide more comprehensive insights into the relationships between sensory patterns and metabolic risk factors across different populations. The authors suggest that upcoming research should implement advanced neuroimaging techniques, such as functional magnetic resonance imaging (fMRI). This involves exposing individuals to targeted sensory stimuli and observing subsequent brain and physiological responses related to metabolic functions, thereby elucidating connections with the development of MetS. Additionally, exploring biochemical responses, and encompassing hormonal and metabolic markers, are suggested in order to provide a comprehensive understanding of the physiological mechanisms involving or underpinning the relationship between sensory patterns and MetS.

## Conclusion

This study examined sensory processing patterns and their association with metabolic risks among community-dwelling people with MetS. Its findings showed that levels of arousal in the proprioceptive and auditory senses were high. Moreover, sensory processing patterns are associated with metabolic risk factors, particularly FBG, DBP, and WC. The findings showed that the FBG levels are associated with sensory preferences in tactile, vestibular, and the proprioceptive senses, the DBP is associated with sensory preferences in vestibular sense, and the WC is associated with sensory arousal in the auditory and smell-taste senses. The findings contribute to raising awareness about the potential impact of sensory processing patterns on metabolic health. Understanding this association can help in designing interventions and for selecting appropriate strategies to reduce the risks and improve overall health and well-being for community-dwelling people with MetS in the future. Moreover, this study provides a foundation for further studies to explore and refine these associations, potentially leading to more targeted and effective interventions.

## Supporting information

**S1 File. The Thai Sensory Patterns Assessment (TSPA).**
(PDF)

**S2 File. STROBE-checklist-cross-sectional studies.**
(PDF)

**S3 File. Data -stress score-metabolic risk variables.**
(PDF)

## Acknowledgments

This research was supported by the Department of Occupational Therapy, Faculty of Associated Medical Sciences, Chiang Mai University.

## Author Contributions

**Conceptualization:** Ilada Pomngen, Tiam Srikhamjak.

**Data curation:** Ilada Pomngen.

**Formal analysis:** Ilada Pomngen.

**Funding acquisition:** Ilada Pomngen.

**Investigation:** Ilada Pomngen.

**Methodology:** Ilada Pomngen, Tiam Srikhamjak.

**Project administration:** Ilada Pomngen.

**Resources:** Ilada Pomngen.

**Software:** Ilada Pomngen.

**Supervision:** Ilada Pomngen, Pornpen Sirisatayawong, Warunee Kumsaiyai, Anuchart Kaunnil, Tiam Srikhamjak.

**Validation:** Ilada Pomngen, Pornpen Sirisatayawong, Tiam Srikhamjak.

**Visualization:** Ilada Pomngen.

**Writing – original draft:** Ilada Pomngen, Anuchart Kaunnil, Tiam Srikhamjak.

**Writing – review & editing:** Ilada Pomngen, Pornpen Sirisatayawong, Warunee Kumsaiyai, Anuchart Kaunnil, Tiam Srikhamjak.

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
