## [Decision Letter · Decision Letter 0]

1 Apr 2024

PONE-D-24-06360Sensory processing patterns as internal factors determining metabolic risk factors among community dwelling people with Metabolic SyndromePLOS ONE

Dear Dr. Pomngen,

Thank you for submitting your manuscript to PLOS ONE. After careful consideration, we feel that it has merit but does not fully meet PLOS ONE’s publication criteria as it currently stands. Therefore, we invite you to submit a revised version of the manuscript that addresses the points raised during the review process.

Two experts in the field have carefully reviewed the manuscript entitled “Sensory processing patterns as internal factors determining metabolic risk factors among community dwelling people with Metabolic Syndrome”. Their comments are appended below.

In light of these reviews and my own reading of the manuscript, I am requesting a major revision and resubmission, in which you will need to respond to each point in each review. Let me focus in some major points that the reviewers and I would like to see addressed. These are: 1) The non-compliance with STROBE guidelines for cross-sectional studies. Note that this is a requirement for publication in Plos ONE (https://journals.plos.org/plosone/s/submission-guidelines#loc-guidelines-for-specific-study-types)2) Both reviewers and I have concerns regarding the TSPA, which we can not access due to language barriers (it is apparently written in Thai language) and the requirement of institutional access to see the relevant documentation. I am requesting that you include the English translation of the TSPA as supplemental material, and clarify on its content, reliability and the rationale behind its use.

3) One reviewer requires more appropriate data analysis and reevaluating the Discussion based on this, and I wish to highlight this requirement as well, together with the recommendation made by the other reviewer, who is an expert in medical sciences, to perform a descriptive analysis grouping variables according to categories such as prediabetes/diabetes, hypertension stage 1/stage 2, etc.  

4) I ask you to do a careful revision of the manuscript in terms of grammar and language. For example, in lines 231-233 you say that "there were high arousal levels...except for propioceptive  and auditory senses", while Table 2 shows the opposite.

There are other points brought out in the reviews and I will carefully attend to your item-by-item responses to them.

We look forward to receiving your revised manuscript.

Kind regards,

Bruno Alejandro Mesz, Ph.D.

Academic Editor

PLOS ONE

3. We note that your Data Availability Statement is currently as follows: [All relevant data are within the manuscript.]

Reviewers' comments:

Reviewer's Responses to Questions

**Comments to the Author**

1. Is the manuscript technically sound, and do the data support the conclusions?

Reviewer #1: No

Reviewer #2: Partly

2. Has the statistical analysis been performed appropriately and rigorously? 

Reviewer #1: No

Reviewer #2: Yes

3. Have the authors made all data underlying the findings in their manuscript fully available?

Reviewer #1: No

Reviewer #2: No

4. Is the manuscript presented in an intelligible fashion and written in standard English?

Reviewer #1: Yes

Reviewer #2: No

5. Review Comments to the Author

Reviewer #1: The authors report on a study on the relationship between preference patterns in different sensory modalities and metabolic risk factors in Thai individuals with metabolic syndrome. The study has the potential to advance our understanding about sensory processing, metabolism, and non-communicable diseases. However, I have some concerns about the statistical analysis and questions about the tools used.

Major concerns

The link between sensory processing and metabolic risk factors is not presented. The authors should clearly explain how there can be a relationship and what might be expected and why.

The rationale and motivation behind the use of the Thai Sensory Profile Assessment Tool is missing, which makes is hard to evaluate the appropriateness of the tool. Why specifically was this tool used? How does it compare to other potential tools? What are the advantages and disadvantages? Has it been used before? Is there a refereed version of this tool? Furthermore, the authors should provide a better description of the tool. For instance, they should explain the rating scales and provide examples of items.

The authors argue they used kernel regression because the data was not normally distributed. However, OLS regression does not make assumptions as to the Gaussian nature of the data but the errors. More tests as to the behavior of the data should be conducted. Using kernel regression does not seem an appropriate method given the high likelihood of omitted variable biases. All the variables from sensory processing patterns data should be included in a single model. There are several ways in which this can be tackled. As a first approximation, OLS with robust standard errors can be used. Furthermore, using generalized linear mixed model and a family that best fits the data may yield better fitting models.

As the authors mentioned, the data is not normally distributed, but they use Spearman’s correlation tests. A correlation test that accounts for non-normality (e.g., Kendall’s Tau). Related to the correlation analysis, even though two sensory pattern variables were found to be significantly correlated with different metabolic risk factors, these correlations were still low.

The Discussion should be reevaluated based on more appropriate statistical analyses.

Reviewer #2: General assessment

This study investigates the potential relationship between sensory processing patterns (sensory preferences and arousals) and metabolic risks in individuals with metabolic syndrome (MetS) (n=117).

I have several suggestions that might improve the manuscript.

While well-written, the manuscript contains grammatical errors requiring thorough revision by authors or an English specialist (ie. Ln68 “Sensory processing refers to ability of…”).

For cross-sectional studies like this, adherence to STROBE guidelines (https://www.strobe-statement.org/) is crucial and I suggest including the checklist indicating the lines where the corresponding items are clearly attended

Also, this is an observational study, so please avoid language implying unsupported causal relationships in all the manuscript (ie. “...internal factors determining metabolic risk factors…”). This editorial might be helpful doi: 10.1111/jan.14311

Finally, I have several concerns related to the Thai Sensory Patterns Assessment-adult version (TSPA), Ref. 15:

Pomngen I, Srikhamjak T, Putthinoi S. Development of the Sensory Patterns Assessment. Chiang Mai: Chiang Mai University; 2020

This appears to be a thesis that probably corresponds to http://cmuir.cmu.ac.th/jspui/handle/6653943832/69493; however, the abstract is in Thai language, probably the full manuscript, and its requires institutional access.

In addition, there is a similar thesis and works with shared authors and links to Google Scholar citations (link for Ref 18 is broken) in the following article

Sutthachai, R.; Kaunnil, A.; Phadsri, S.; Pomngen, I.; Stanley, M.; Srikhamjak, T. Development of Thai Sensory Patterns Assessment Tool for Children Aged 3–12 Years: Caregiver-Version. Healthcare 2022, 10, 1968. https://doi.org/10.3390/healthcare10101968 :

Ref 15: Srikhamjak, T.; Sawlom, S.; Munkhetvit, P.; Apikimonkon, H. Thai Sensory Profile Assessment Tool; Chiang Mai University: Chiang Mai, Thailand, 2007. (In Thai)

Ref 16: Pomngen, I. Development of the Sensory Patterns Assessment. Master’s Thesis, Chiang Mai University, Chiang Mai, Thailand, 24 February 2020

Ref 17: Pomngen, I.; Srikhamjak, T.; Putthinoi, S. Development of Thai’s Sensory Patterns Assessment Tool for Adolescents and Adults. J. Health Sci. Res. 2020, 14, 76–85

Tracked here: https://he01.tci-thaijo.org/index.php/JHR/article/view/228407/164593 (In Thai)

Ref 18: Srikamjak, T.; Saolarm, S.; Munkhetvit, P.; Apikonekorn, H. The sensory profile assessment tool: TSPA. In Proceedings of the Opening Word Optimizing Occupational Therapy, The 5th Asia Pacific Occupational Therapy Congress, Chiang Mai, Thailand, 19–24 November 2011; Volume A0193, p. 183

Ref 19: Kanchanawong, T.; Prasoetsang, T.; Limvongvatana, F.; Ooraikul, L.; Kanupan, S.; Srikamjak, T.; Gomutbutra, P. The Feasibility of the Thai Sensory Profile Assessment Tool (TSPA) for Classifying the participants for Mind-Body intervention. medRxiv 2022, 1–15

Tracked here: https://www.medrxiv.org/content/10.1101/2022.06.05.22275861v2.full

A preprint was withdrawn for being “enormous modified”, probably the same as Ref 19 https://osf.io/preprints/psyarxiv/726qr

Found here: https://scholar.archive.org/work/rdcu4j7mfnghrkeef3nt5zod2i/access/wayback/https://files.osf.io/v1/resources/726qr/providers/osfstorage/622d0c5953a4e806b7516047?action=download&direct&version=2

The authors indicate that the Thai Sensory Profile Assessment Tool (TSPA) “...can be accessed via the website www.tspatools.com”, but the link is broken.

In this case the authors must include the English translation for the Thai Sensory Patterns Assessment-adult version (TSPA) as supplemental material, and clarify in its content, constructs, reliability, administration and scoring procedures, results from factor analysis for the assessment of the intended dimensions in the questionnaire.

Title

I suggest a simplification of the title and avoiding causal language such as “determining”, as the study (cross-sectional / correlational) was not intended to identify causality but to explore potential relationships ie. “Relationships between sensory processing patterns and metabolic risk factors”

Abstract

Background: Metabolic syndrome is a non-communicable disease, please clarify.

Methods: The authors must clarify data categorization and analysis procedures, specifying the variables used and their objectives

Conclusion: What were the most relevant study findings? and optionally incorporate perspectives or future directions.

Keywords: I suggest that the suggested keywords be different from those used in the Title to expand the visibility of the work

Introduction

Ln 58-60, metabolic syndrome is also a non-communicable disease, please clarify

Ln 70-72, As this is central to the study, I suggest that the authors expand the definition of “sensory processing patterns”, including how it is measured (ie. Are there different instruments for its assessment? With the same sensory domains?), what are the possible resulting categories, and how can the results be interpreted. Also, this information might be useful for discussing the results by contrasting these tools with the one used in this study (Thai Sensory Patterns Assessment-adult, TSPA).

Ln 74-76, sensory processing patterns might be related to the development of chronic diseases including MetS, but the authors have yet to present a strong and specific argument to support that possibility

Ln77-86 Please clarify if the studies used the TSPA or other test(s) for sensory processing patterns. If there are individuals classified with “high sensory sensitivity”, are there those with low or moderate sensory sensitivities? What are the characteristics of these groups?

Ln92, please clarify how limited is the direct evidence on the relationship between sensory processing patterns and MetS/metabolic risks, and how this has been explored.

Ln 97–99, please include specific studies and references.

Ln100-106, in addition to my concerns outlined in the general assessment, please provide clarification in this section regarding the development, domains, usage and validity of the Thai Sensory Patterns Assessment-adult (TSPA) as well as how it has been used previously. Given that REF 15 is a thesis written in Thai language with restricted access to institutional members, it is essential that the authors make the questionnaire available in English as supplemental material.

Ln111-115, while acknowledging the importance of this aspect, the design of this study does not facilitate the assessment of the influence of sensory processing patterns in the development and progression of metabolic syndrome (MetS) and its associated factors. Please rephrase.

Ln120-125, this section is about perspectives. However, I suggest that the authors clarify the hypothesis being explored and expected outcomes based on the information presented in Introduction. For example, it would be beneficial to specify which sensory processing pattern(s) are expected to be associated with the metabolic risk variables: fasting blood glucose (FBG), systolic blood pressure (SBP) and diastolic blood pressure (DBP), and/or waist circumference (WC). By elucidating these relationships in the Introduction, the rationale behind the hypothesis may become clearer.

Methods

Ln143-144, I am not familiar with the demographics of Thailand, but the current wording may imply that only 117 individuals with MetS resided in the Namprea sub-district. Please confirm if this is accurate? Also, is it the same Namprea than Nam Phrae?

Ln153, please elaborate on the impact of the level of education and how it was implemented in this study. In addition, I am not familiar with the Mental State Examination T10 (MSET-10). Please provide clarification of this tool and why it was selected to screen cognitive impairments. Also, include a relevant and appropriate reference for further context and insight.

Ln158, I am not familiar with purposive sampling, so it would be helpful to give a concise overview, the rationale behind its use, and the specific criteria used to fulfill that purpose. Furthermore, please incorporate this information into the discussion of the study’s limitations, particularly regarding its ability to generalize findings to the population.

Ln160 Please clarify the screening and identification process.

Ln161-163, previously it was mentioned 117 participants, please clarify and also the current wording may imply that only 145 individuals with MetS resided in the community. If that is the case, why was there not a direct invitation instead of “...flyers and word-of-mouth”?

Ln168 wasn’t the demographic questionnaire already completed along with the MSET-10? (Ln165). Also, where are the results of the MSET-10?

Ln185, please, include the English version of the TSPA

Ln189, why are smell and taste combined?

Ln190 please provide the reference for the factor analysis conducted for TSPA

Ln195-197 Given language institutional access barriers to the referred thesis (Ref 15), the authors must provide the English version of the Thai Sensory Patterns Assessment-adult version (TSPA). In addition, it would be beneficial to provide a translation of the referenced thesis for verification and reproducibility purposes, ie. the meaning of IOC/ICC and parts I and II.

Analysis

I recommend conducting a comprehensive descriptive analysis, which should involve grouping variables as follows:

Blood pressure (BP): participants with hypertension could be categorized into Stage 1 (130-139 / 80-89) and Stage 2 (≥ 140 / ≥ 90)

Fasting blood glucose (FBG): Prediabetes (100-125 mg/dL) and diabetes (≥ 126 mg/dL)

Waist circumference (WC): For example, 80-85cm and >85 (for women); 90-95 and >95 (for men)

Are there any differences by sex or age?

Ln208-210 Please, clarify how this was implemented

Ln210-211 If you are referring to p-values, these are not just probsbilities so I suggest to rephrase: "p-values below 0.05 were considered statistically significant."

Results

Table 1

Ln225, Please, include the total participants in the title

I suggest to modify “Gender” to “Sex” (as suggested in: 10.1001/jama.2016.16405)

There is no need of “Postgraduate” category as there are no individuals in that group

Please, verify the data in “Household income” as there are reported only 116 individuals

Also, include the values for metabolic risks

Table 2.

Please clarify “Results” as well as low, moderate and high

Ln239, the results from this section should be explained in Discussion section

Table 3.

Please clarify the test performed to calculate p-values

Ln241-242 How is this result explained? For the Discussion section

Ln255 Please clarify here the test used to calculate p-values, rather than in the title of the table

Ln257, rephrase “Predicting” and remove causal language in all the manuscript and tables

Table 4

I suggest caution and to take into account limitations regarding prediction purposes, especially when assumptions, model complexity, and evaluation metrics are not clearly reported. It would be more appropriate to use terms like “association” or “relationship” instead of “prediction” in this context

Discussion

Overall, this section requires revision for clarity and to maintain focus on metabolic-related factors rather than “stress”. Speculation should be avoided, and the true impact of the findings should be accurately assessed

Ln273, I suggest that the authors compare their results with what has been previously studied, including how it was studied and specifying the sensory modalities involved.

Ln302 What are “sensory sensitivities”, “heightened sensory sensitivity” (Ln312), “sensory sensitivity” (Ln414)?

Conclusions

Please be specific on which sensory patters were associated with metabolic risk factors. Omit any redundant mentions of limitations in this section, as they have already been addressed previously

Author contributions

Ln454 There is no visualization in this manuscript, please clarify.

6. PLOS authors have the option to publish the peer review history of their article (what does this mean?). If published, this will include your full peer review and any attached files.

Reviewer #1: No

Reviewer #2: No

---

## [Author Response · Author response to Decision Letter 0]

13 May 2024

Response to Reviewer #1:

We greatly appreciate your thoughtful comments concerning our manuscript "Sensory processing patterns as internal factors determining metabolic risk factors among community dwelling people with Metabolic Syndrome" (Manuscript ID: PONE-D-24-06360). Those comments are all valuable and very helpful for revising and improving our paper. We have studied the comments carefully and have made correction which we hope meet your approval. Revised portion are marked in yellow highlighting in the paper and and indicated by blue text.

The major concerns in the paper and the responds to the reviewer’s comments are as flows:

Point 1: The link between sensory processing and metabolic risk factors is not presented. The authors should clearly explain how there can be a relationship and what might be expected and why:

Response 1: We appreciate your comments and suggestions. We have addressed the potential link between sensory processing and metabolic risk factors in the Introduction section, specifically on pages 4-5, lines 85-101.

In summary, Sensory processing patterns influence human emotions and behaviors, including what we eat, what make us feel stressful. Previous studies have shown associations between emotions and behaviors relevant to Metabolic Syndrome (such as eating habits, physical inactivity, and stress) and sensory processing patterns. However, these relationships have been observed in other groups, such as healthy workers, adults, adolescents with Type 1 diabetes, and individuals with multiple sclerosis.

Furthermore, previous research has indicated a relationship between sensory processing patterns and metabolic risk variables, such as BMI. However, there are currently no studies focusing on individuals with MetS or exploring other metabolic risk variables, such as blood pressure, fasting blood glucose, and waist circumference. Therefore, we anticipate finding a relationship between sensory processing patterns and metabolic risk factors, which serve as diagnostic criteria for MetS.

Point 2: The rationale and motivation behind the use of the Thai Sensory Profile Assessment Tool is missing, which makes is hard to evaluate the appropriateness of the tool. Why specifically was this tool used? How does it compare to other potential tools? What are the advantages and disadvantages? Has it been used before? Is there a refereed version of this tool? Furthermore, the authors should provide a better description of the tool. For instance, they should explain the rating scales and provide examples of items. 

Response 2: We appreciate and agree with your comments and suggestions regarding the Thai Sensory Profile Assessment Tool (TSPA). We have now included additional information about the TSPA to clarify the rationale and motivation behind its use. This includes comparisons to other potential tools, discussion of its advantages and disadvantages, previous research studies that have utilized the TSPA, an explanation of the rating scales and examples of items as well as other important information. This information can be found in the Introduction section on pages 6-7, lines 132-168, and in the Instruments section on pages 12-13, lines 267-305.

Point 3: The authors argue they used kernel regression because the data was not normally distributed. However, OLS regression does not make assumptions as to the Gaussian nature of the data but the errors. More tests as to the behavior of the data should be conducted. Using kernel regression does not seem an appropriate method given the high likelihood of omitted variable biases. All the variables from sensory processing patterns data should be included in a single model. There are several ways in which this can be tackled. As a first approximation, OLS with robust standard errors can be used. Furthermore, using generalized linear mixed model and a family that best fits the data may yield better fitting models.

Response 3: We appreciate the reviewer's helpful suggestion regarding an appropriate statistical analysis method. We have now made a change from kernel regression to a generalized linear mixed model, as suggested, on page 14, lines 322-328. Consequently, the results have also been updated on pages 18-19, lines 382-393.

Point 4: As the authors mentioned, the data is not normally distributed, but they use Spearman’s correlation tests. A correlation test that accounts for non-normality (e.g., Kendall’s Tau). Related to the correlation analysis, even though two sensory pattern variables were found to be significantly correlated with different metabolic risk factors, these correlations were still low.

Response 4: We truly appreciate your comments regarding the need for more appropriate statistical analyses. Following your suggestion, we have now changed the statistical analysis method from Spearman correlation to Kendall’s tau, on page 13, line 319. Consequently, the results have also been updated on pages 17-18, lines 371, 375, 377-380. Moreover, in your concern that the correlations were still low, we have acknowledged the limitation of the study in the limitation section, specifically regarding the low correlations found. This limitation highlights the caution needed when interpreting and applying these findings in practice on pages 24, lines 513-516, 526-528.

Point 5: The Discussion should be reevaluated based on more appropriate statistical analyses.

Response 5: We would like to thank you for your comments and suggestion. We have now changed the statistical analyses medthod based on your suggestion (from Spearman correlation to Kendall’s tau and from Kernel regression to General Linear Mixed Model, both of which are more appropriate for non-parametric or non- normally distributed data), and we also revised and ubdated the discussion section. Now, it is on the page 19-24, line 394-524. 

Special thanks to you for your good comments and suggestion. We tried our best to improve the manuscript and made some changes in the manuscript. 

We appreciate for reviewers’ warm work earnestly and hope that the correction will meet with approval. Once again, thank you very much indeed for your comments and suggestions.

Best regards,

Ilada Pomngen and research team

Response to Reviewer #2:

We greatly appreciate your thoughtful comments concerning our manuscript "Sensory processing patterns as internal factors determining metabolic risk factors among community dwelling people with Metabolic Syndrome" (Manuscript ID: PONE-D-24-06360). Those comments are all valuable and very helpful for revising and improving our paper. We have studied the comments carefully and have made correction which we hope meet your approval. Revised portion are marked in yellow highlighting in the paper and indicated by blue text.

The major concerns in the paper and the responds to the reviewer’s comments are as flows:

Point 1: While well-written, the manuscript contains grammatical errors requiring thorough revision by authors or an English specialist (ie. Ln68 “Sensory processing refers to ability of…”).

Response 1: We would like to apologize for this mistake and thank you for your comments and suggestion about the grammatical errors. After we have made revision carefully based on your comments and suggestion, we already sent the manuscript to an English specialist, who is a native English, for revision. 

Point 2: For cross-sectional studies like this, adherence to STROBE guidelines (https://www.strobe-statement.org/) is crucial and I suggest including the checklist indicating the lines where the corresponding items are clearly attended. 

Response 2: We are thankful with your comments and suggestion regarding the non-compliance with STROBE guidelines for cross-sectional studies. We have carefully reviewed our manuscript to ensure it meets the requirements for publication by utilizing the STROBE guidelines for cross-sectional studies. Additionally, we have included the importan information required from the STROBE guidelines in our manuscript, and included a checklist indicating the lines where the corresponding items are clearly addressed and have uploaded this checklist as supplemental material.

Point 3: Also, this is an observational study, so please avoid language implying unsupported causal relationships in all the manuscript (ie. “...internal factors determining metabolic risk factors…”). This editorial might be helpful doi: 10.1111/jan.14311.

Response 3: We thank the reviewer for this useful suggestion about avoiding causal language. We have now addressed it in the title of the manuscript on page 1 by using a simplified version: 'Relationships between sensory processing patterns and metabolic risk factors among community dwelling people with Metabolic Syndrome’, on page 1, lines 1-3.

Point 4: Finally, I have several concerns related to the Thai Sensory Patterns Assessment-adult version (TSPA), Ref. 15:

Pomngen I, Srikhamjak T, Putthinoi S. Development of the Sensory Patterns Assessment. Chiang Mai: Chiang Mai University; 2020.

This appears to be a thesis that probably corresponds to http://cmuir.cmu.ac.th/jspui/handle/6653 943832/ 69493; however, the abstract is in Thai language, probably the full manuscript, and its requires institutional access. 

In addition, there is a similar thesis and works with shared authors and links to Google Scholar citations (link for Ref 18 is broken) in the following article

Sutthachai, R.; Kaunnil, A.; Phadsri, S.; Pomngen, I.; Stanley, M.; Srikhamjak, T. Development of Thai Sensory Patterns Assessment Tool for Children Aged 3–12 Years: Caregiver-Version. Healthcare 2022, 10, 1968. https://doi.org/10.3390/healthcare10101968.

Ref 15: Srikhamjak, T.; Sawlom, S.; Munkhetvit, P.; Apikimonkon, H. Thai Sensory Profile Assessment Tool; Chiang Mai University: Chiang Mai, Thailand, 2007. (In Thai)

Ref 16: Pomngen, I. Development of the Sensory Patterns Assessment. Master’s Thesis, Chiang Mai University, Chiang Mai, Thailand, 24 February 2020 

Ref 17: Pomngen, I.; Srikhamjak, T.; Putthinoi, S. Development of Thai’s Sensory Patterns Assessment Tool for Adolescents and Adults. J. Health Sci. Res. 2020, 14, 76–85

Tracked here: https://he01.tci-thaijo.org/index.php/JHR/article/view/228407/164593 (In Thai)

Ref 18: Srikamjak, T.; Saolarm, S.; Munkhetvit, P.; Apikonekorn, H. The sensory profile assessment tool: TSPA. In Proceedings of the Opening Word Optimizing Occupational Therapy, The 5th Asia Pacific Occupational Therapy Congress, Chiang Mai, Thailand, 19–24 November 2011; Volume A0193, p. 183

Ref 19: Kanchanawong, T.; Prasoetsang, T.; Limvongvatana, F.; Ooraikul, L.; Kanupan, S.; Srikamjak, T.; Gomutbutra, P. The Feasibility of the Thai Sensory Profile Assessment Tool (TSPA) for Classifying the participants for Mind-Body intervention. medRxiv 2022, 1–15

Tracked here: https://www.medrxiv.org/content/10.1101/2022.06.05.22275861v2.full

A preprint was withdrawn for being “enormous modified”, probably the same as Ref 19 https://osf.io/preprints/psyarxiv/726qr

Found here: https://scholar.archive.org/work/rdcu4j7mfnghrkeef3nt5zod2i/access/wayback/https://files.osf.io/v1/resources/726qr/providers/osfstorage/622d0c5953a4e806b7516047?action=download&direct&version=2

The authors indicate that the Thai Sensory Profile Assessment Tool (TSPA) “...can be accessed via the website www.tspatools.com”, but the link is broken.

In this case the authors must include the English translation for the Thai Sensory Patterns Assessment-adult version (TSPA) as supplemental material, and clarify in its content, constructs, reliability, administration and scoring procedures, results from factor analysis for the assessment of the intended dimensions in the questionnaire.

Response 4: We truly appreciate your comments and suggestions regarding the Thai Sensory Patterns Assessment-adult version (TSPA), Ref. 15. In response, we have included the English translation of the Thai Sensory Patterns Assessment-adult version (TSPA), as well as the thesis abstract in English version of Ref. 15. Additionally, we have provided related information to clarify its content, constructs, validity, reliability, administration, and scoring procedures. These details have been included as supplemental material.

Point 5: Title I suggest a simplification of the title and avoiding causal language such as “determining”, as the study (cross-sectional / correlational) was not intended to identify causality but to explore potential relationships ie. “Relationships between sensory processing patterns and metabolic risk factors”.

Response 5: We thank the reviewer for this useful suggestion about avoiding causal language. We have now addressed it in the title of the manuscript on page 1 by using a simplified version: 'Relationships between sensory processing patterns and metabolic risk factors among community dwelling people with Metabolic Syndrome’, on page 1, lines 1-3.

Point 6: Abstract Background: Metabolic syndrome is a non-communicable disease, please clarify.

Response 6: We would like to apologize for this mistake and thank you for your comments and suggestion. We have now rephased the sentence for make it clearer that “Metabolic Syndrome (MetS) is a cluster of metabolic risk factors that increases the risk of other serious health problems, particularly cardiovascular diseases and stroke”, on page 2, lines 27-28.

Point 7: Abstract Methods: The authors must clarify data categorization and analysis procedures, specifying the variables used and their objectives.

Response 7: We appreciate and would like to thank you for your comments and suggestions. We have clarified and specified the variables used (metabolic risk variables and sensory processing patterns in six sensory modalities) and their objectives for examining the relationship on page 2, lines 35-42. However, due to the word limit (300 words), we attempted to include the required information. If you have any further suggestions, please let us know.

Point 8: Abstract Conclusion: What were the most relevant study findings? and optionally incorporate perspectives or future directions.

Response 8: We. We appreciate and would like to thank you for your comments and suggestions. We have clarified the study findings and provided suggestions for future research. This information is now included on page 3, lines 52-54.

Point 9: Abstract Keywords: I suggest that the suggested keywords be different from those used in the Title to expand the visibility of the work.

Response 9: We appreciate the reviewer's useful suggestion. To expand the visibility of our work, we have added some keywords that are different from those used in the title, such as 'association', Correlation’ and ‘Metabolic health’. This addition can help potential readers better understand the focus of our study. The keywords are now included on page 3, lines 57."

Point 10: Introduction Ln 58-60, metabolic syndrome is also a non-communicable disease, please clarify 

Response 10: We would like to apologize for this mistake and thank you for your comments and suggestion. We have now rephased the sentence for make it clearer ‘Currently, one of the main global public-health challenges is metabolic syndrome (MetS), which is defined as a cluster of metabolic abnormalities, such as insulin resistance, visceral obesity, hypertension, and dyslipidemia [2]. Following the occurrence of MetS, people are more likely to be exposed to and develop other serious health problems, in particular cardiovascular diseases[3] and stroke [4, 5].’, on page 3, lines 61-65.

Point 11: Ln 70-72, As this is central to the study, I suggest that the authors expand the definition of “sensory processing patterns”, including how it is measured (ie. Are there different instruments for its assessment? With the same sensory domains?), what are the possible resulting categories, and how can the results be interpreted. Also, this information might be useful for discussing the results by contrasting these tools with the one used in this study (Thai Sensory Patterns Assessment-adult, TSPA).

Response 11: We appreciate the reviewer's useful suggestion. To enhance clarity, we have expanded the definition of 'sensory processing patterns' on page 4, in lines 74-84. 

---

## [Decision Letter · Decision Letter 1]

5 Jul 2024

PONE-D-24-06360R1Relationships between sensory processing patterns and metabolic risk factors among community dwelling people with Metabolic Syndrome: A cross-sectional and correlational research designPLOS ONE

Dear Dr. Pomngen,

Thank you for submitting your manuscript to PLOS ONE. After careful consideration, we feel that it has merit but does not fully meet PLOS ONE’s publication criteria as it currently stands. Therefore, we invite you to submit a revised version of the manuscript that addresses the points raised during the review process.

Thank you for submitting your revision of your article “Relationships between sensory processing patterns and metabolic risk factors among community dwelling people with Metabolic Syndrome: A cross-sectional and correlational research design” I have read your revised manuscript, with the last round of reviewers’

comments in hand. Thank you for your careful attention to these corrections,

additions, and clarifications. I will not send your revision back to them for another reading, but I am requesting you to address the following points (see the reviews below).

 The GLMM needs work. Please report the family and link function used in the model and why. Report the model fit statistics, including the null models. Please report the software used to conduct the analyses.  Report the specific test used for the power calculation in G*Power.  Be careful not to overstretch the results, given that the correlations you obtained are weak, for example when you speak of “unexpected results” on relationship between WC and auditory arousal.Please revise the “other comments” at the end of  review 1 and the minor points made by reviewer 2.Please do a new English check. I have seen the language check document you attached, but new mistakes appeared, such as “The example of items…” (line 289), “For psychometric properties. The…”(line 297), line 304 should be 0.849,  lines 408 to 411, and several others. Avoid anthropomorphizing, as reviewer 1 says, for example “Kendall’s Tau found” in the abstract.Line 440: should it be tactile sense?  

We look forward to receiving your revised manuscript.

Kind regards,

Bruno Alejandro Mesz, Ph.D.

Academic Editor

PLOS ONE

Reviewers' comments:

Reviewer's Responses to Questions

**Comments to the Author**

1. If the authors have adequately addressed your comments raised in a previous round of review and you feel that this manuscript is now acceptable for publication, you may indicate that here to bypass the “Comments to the Author” section, enter your conflict of interest statement in the “Confidential to Editor” section, and submit your "Accept" recommendation.

Reviewer #1: (No Response)

Reviewer #2: All comments have been addressed

2. Is the manuscript technically sound, and do the data support the conclusions?

Reviewer #1: Partly

Reviewer #2: Yes

3. Has the statistical analysis been performed appropriately and rigorously? 

Reviewer #1: No

Reviewer #2: Yes

4. Have the authors made all data underlying the findings in their manuscript fully available?

Reviewer #1: No

Reviewer #2: Yes

5. Is the manuscript presented in an intelligible fashion and written in standard English?

Reviewer #1: No

Reviewer #2: Yes

6. Review Comments to the Author

Reviewer #1: The authors have improved the manuscript. However, several issues still require attention. Overall most of the discussion and conclusions do not seem to be well supported by the results. Furthermore, the statistical analysis needs to be reported with more clarity and transparency. As of now, the specific GLMM used was not disclosed.

Major comments

The abstract is too long and convoluted. Please make the abstract more succinct. Briefly state the background, motivation, methodology, and findings. Do not report statistics.

Report the specific test used for the power calculation in G*Power.

There is basically no correlation between any of the variables. The only couple that are significant are too low. The authors are overinterpreting this correlations.

The GLMM needs work. Please report the family and link function used in the model and why. Report the model fit statistics, including the null models. Please report the software used to conduct the analyses.

The discussion on the link between auditory sensory preference and WC given that this is based only on a (low) correlation. In addition, the GLMM showed no significant effect.

The discussion on tactile sensory preference needs some work. There is no direction in the preference for food texture. Are all food textures disliked? Or are specific textures dislike, which drives dislike to specific kinds of foods.

Other comments

L.75, what is body stimuli?

Please capitalize COVID-19

What is behaviorally observing methods?

L.315, what does percentage mean? Percentage of what?

L.382, "the general linear mixed model discovered". Please avoid anthropomorphism here. The analysis did not "discover", it may have revealed.

Reviewer #2: The authors attended my suggestions. It was fundamental to include an English version of Thai Sensory Patterns Assessment-adult version (TSPA) as well as the adherence to STROBE guidelines (in text and with a supplementary checklist). The descriptive analysis provides an in-depth exploration to the data and the readability has been improved. The manuscript can be accepted

I have minor suggestions after which the manuscript can be published:

Ln 171: I suggest removing the parenthesis in "pattern(s)".

Ln 281: Please modify to "confirmatory factor analysis (CFA)".

Ln 297: Please modify to "For psychometric properties, the content...".

Ln 350: Please modify to "p<0.001". Correct throughout the document, including tables. Although the software used generates an output of 0.000 or similar, this does not mean the result is literally zero, but rather a problem with the software displaying the true value. Hence, to avoid confusion, this should be converted, i.e., from p = 0.000 to p < 0.001.

Ln 408-411 This section requires rephrasing for clarity

Ln 578 There is a typo in “A,nuchart Kaunnil”

7. PLOS authors have the option to publish the peer review history of their article (what does this mean?). If published, this will include your full peer review and any attached files.

Reviewer #1: No

Reviewer #2: No

---

## [Author Response · Author response to Decision Letter 1]

16 Jul 2024

Response (round-2) to editor

We greatly appreciate your thoughtful comments concerning our manuscript entitled "Sensory processing patterns as internal factors determining metabolic risk factors among community dwelling people with Metabolic Syndrome" (Manuscript ID: PONE-D-24-06360R1). Those comments are all valuable and very helpful for revising and improving our paper. We have studied comments carefully and have made correction which we hope meet with approval. Revised portion are marked in yellow highlighting in the paper and indicated by blue text.

The major points in the paper and the responds to the reviewers and editor comments are as flowing:

Point 1: The GLMM needs work. Please report the family and link function used in the model and why. Report the model fit statistics, including the null models. Please report the software used to conduct the analyses.

Response 1: We truly appreciate your comments regarding the GLMM. Following your suggestion, we have now added the information regarding the GLMM analysis, including the family, link function, model fit statistics and software used to conduct the analyses. Now, the revision is on page 14, lines 324-331.

Point 2: Report the specific test used for the power calculation in G*Power

Response 2: We appreciate on your comments and suggestions. We have added the specific test for the power calculation in G*Power on page 8, lines 194-195.

Point 3: Be careful not to overstretch the results, given that the correlations you obtained are weak, for example when you speak of “unexpected results” on relationship between WC and auditory arousal.

Response 3: We would like to thank you for your comments and suggestion. We have carefully revised the manuscript and rephased this part, ensuring not to overstretch the interpretation of the results. Now, it is on the page 22, line 473-478.

Point 4: Please revise the “other comments” at the end of review 1 and the minor points made by reviewer 2.

Response 4: We truly appreciate and agree with your and reviewers’ comments. We have carefully revised all of the feedback and suggestions from both reviewers. 

Point 5: Please do a new English check. I have seen the language check document you attached, but new mistakes appeared, such as “The example of items…” (line 289), “For psychometric properties. The…” (line 297), line 304 should be 0.849, lines 408 to 411, and several others. Avoid anthropomorphizing, as reviewer 1 says, for example “Kendall’s Tau found” in the abstract.

Response 5: We truly appreciate and agree with your comments regarding the manuscript’s grammar and language. After carefully revising the manuscript based on your feedback and the suggestions of the two reviewers, we have sent it again to an English specialist who is a native English speaker for further revision. Additionally, we have a copy of the manuscript showing track changes by a native English as supporting information.

- “The examples of items…” (line 284)

- “For psychometric properties. The…” (line 292)

- “0.849” (line 299)

- line 412-414

- the abstract (line 38 and 42)

Point 6: Line 440: should it be tactile sense?

Response 6: Thank you for your comment. We have reviewed the manuscript and confirmed that the use of the term "tactile sense" in line 440 is correct, and this finding showed in Table 5, which analyzed by the GLMM. However, to avoid any confusion, we have added an explanation in the manuscript referencing the specific statistics that support the significance to provide clearer understanding for the readers. Now, the revision is on page 21, line 445.

Special thanks to you for your good comments and suggestion. We tried our best to improve the manuscript and made some changes in the manuscript. 

We appreciate for editor and reviewers’ warm work earnestly and hope that the correction will meet with approval. Once again, thank you very much indeed for your comments and suggestions.

Best regards,

Ilada Pomngen and research team

Response to Reviewer (Round-2) #1:

We greatly appreciate your thoughtful comments concerning our manuscript "Sensory processing patterns as internal factors determining metabolic risk factors among community dwelling people with Metabolic Syndrome" (Manuscript ID: PONE-D-24-06360R1). Those comments are all valuable and very helpful for revising and improving our paper. We have studied the comments carefully and have made correction which we hope meet your approval. Revised portion are marked in yellow highlighting in the paper and and indicated by blue text.

The major concerns in the paper and the responds to the reviewer’s comments are as flows:

Point 1: The abstract is too long and convoluted. Please make the abstract more succinct. Briefly state the background, motivation, methodology, and findings. Do not report statistics.

Response 1: Thank you for your valuable feedback. We have revised the abstract to be more succinct without the report of statistics, while retaining the information that another reviewer was concerned about. This revision is on page 2, lines 26-48.

Point 2: Report the specific test used for the power calculation in G*Power.

Response 2: We appreciate on your comments and suggestions. We have added the specific test for the power calculation in G*Power on page 8, lines 194-195.

Point 3: There is basically no correlation between any of the variables. The only couple that are significant are too low. The authors are overinterpreting this correlations.

Response 3: We appreciate the reviewer's helpful suggestion. We have carefully revised the manuscript, ensuring not to overstretch the interpretation of the results. We have added the sentence to inform the readers to pay attention to the low correlations observed on page 21, lines 441-444. Moreover, in your concern that the correlations were still low, we have acknowledged the limitation of the study in the limitation section, specifically regarding the low correlations found. This limitation highlights the caution needed when interpreting and applying these findings in practice on page 24, lines 518-522; page 25, lines 532-534.

Point 4: The GLMM needs work. Please report the family and link function used in the model and why. Report the model fit statistics, including the null models. Please report the software used to conduct the analyses.

Response 4: We truly appreciate your comments regarding the GLMM. Following your suggestion, we have now added the information regaeding the GLMM analysis, including the family, link function, medel fit statistics and software used to conduct the analyses, on page 14, lines 324-331. 

Point 5: The discussion on the link between auditory sensory arousal and WC given that this is based only on a (low) correlation. In addition, the GLMM showed no significant effect.

Response 5: We would like to thank you for your comments and suggestion. We have carefully revised the manuscript, ensuring not to overstretch the interpretation of the results. Now, it is on the page 22, lines 473-478.

Point 6: The discussion on tactile sensory preference needs some work. There is no direction in the preference for food texture. Are all food textures disliked? Or are specific textures dislike, which drives dislike to specific kinds of foods.

Response 6: Thank you for your insightful comment. We have revised the section to clarify the relationship between sensory sensitivity and food texture preferences. The revised sentence now specifies that individuals with low preference for certain tactile sensations tend to dislike specific textures, which often leads to the avoidance of foods with those textures. Now, it is on the page 23, line 487-491. 

Point 7: L.75, what is body stimuli?

Response 7: We would like to apologize and thank you for your comments and suggestion. We have now corrected it to be internal body stimuli. Now, it is on the page 3, line 69. 

Point 8: Please capitalize COVID-19

Response 8: We would like to thank you for your comments and suggestion. We have now correted it. Now, it is on the page 7, line 161. 

Point 9: What is behaviorally observing methods?

Response 9: We would like to thank you for your comments and suggestion. We have modified it to make it clearer. Now, it is on the page 6, line 128-129. 

Point 10: L.315, what does percentage mean? Percentage of what?

Response 10: We appreciate the reviewer's helpful suggestion regarding an appropriate statistical analysis method. We have now made a change by incorporating a more detailed description of the descriptive analyses. This revision is on page 13, lines 310-313.

Point 11: L.382, "the general linear mixed model discovered". Please avoid anthropomorphism here. The analysis did not "discover", it may have revealed.

Response 11: We would like to apologize and thank you for your comments and suggestion. We have now revised it. Now, it is on the page 18, line 386. 

Special thanks to you for your good comments and suggestion. We tried our best to improve the manuscript and made some changes in the manuscript. 

We appreciate for reviewers’ warm work earnestly and hope that the correction will meet with approval. Once again, thank you very much indeed for your comments and suggestions.

Best regards,

Ilada Pomngen and research team

Response (round-2) to Reviewer #2:

We greatly appreciate your thoughtful comments concerning our manuscript "Sensory processing patterns as internal factors determining metabolic risk factors among community dwelling people with Metabolic Syndrome" (Manuscript ID: PONE-D-24-06360R1). Those comments are all valuable and very helpful for revising and improving our paper. We have studied the comments carefully and have made correction which we hope meet your approval. Revised portion are marked in yellow highlighting in the paper and indicated by blue text.

The major concerns in the paper and the responds to the reviewer’s comments are as flows:

Point 1: Ln 171: I suggest removing the parenthesis in "pattern(s)".

Response 1: We would like to apologize for this mistake and thank you for your comments. Based on your comments and suggestion, we have already checked and corrected it, on page 7, lines 165-166.

Point 2: Ln 281: Please modify to "confirmatory factor analysis (CFA)".

Response 2: We would like to apologize for this mistake and thank you for your comments and suggestion. We have now checked and corrected it, on page 12, line 277.

Point 3: Ln 297: Please modify to "For psychometric properties, the content...".

Response 3: We would like to apologize for this mistake and thank you for your comments and suggestion. We have now checked and corrected it, on page 12, line 292.

Point 4: Ln 350: Please modify to "p<0.001". Correct throughout the document, including tables. Although the software used generates an output of 0.000 or similar, this does not mean the result is literally zero, but rather a problem with the software displaying the true value. Hence, to avoid confusion, this should be converted, i.e., from p = 0.000 to p < 0.001.

Response 4: We truly appreciate your comments and useful suggestions. We have carefully checked and correted them, on page 15, line 354; page 16, Table 2; page 18-19, Table 5.

Point 5: Ln 408-411 This section requires rephrasing for clarity

Response 5: We thank the reviewer for this useful suggestion. We have now rephrased this section, on page 20, lines 412-414.

Point 6: Ln 578 There is a typo in “A,nuchart Kaunnil”

Response 6: We would like to apologize for this mistake and thank you for your comments and suggestion. We have now checked the typo and corrected, on page 27, line 585.

Special thanks to you for your good comments and suggestion. We tried our best to improve the manuscript and made some changes in the manuscript. 

We appreciate for reviewers’ warm work earnestly and hope that the correction will meet with approval. Once again, thank you very much indeed for your comments and suggestions.

Best regards,

Ilada Pomngen and research team

---

## [Editor Report · Decision Letter 2]

23 Jul 2024

Relationships between sensory processing patterns and metabolic risk factors among community dwelling people with Metabolic Syndrome: A cross-sectional and correlational research design

PONE-D-24-06360R2

Dear Dr. Pomngen,

We’re pleased to inform you that your manuscript has been judged scientifically suitable for publication and will be formally accepted for publication once it meets all outstanding technical requirements.

Kind regards,

Bruno Alejandro Mesz, Ph.D.

Academic Editor

PLOS ONE

---

## [Editor Report · Acceptance letter]

25 Jul 2024

PONE-D-24-06360R2 

PLOS ONE

Dear Dr. Pomngen, 

I'm pleased to inform you that your manuscript has been deemed suitable for publication in PLOS ONE. Congratulations! Your manuscript is now being handed over to our production team.

Kind regards, 

on behalf of

Dr. Bruno Alejandro Mesz 

Academic Editor

PLOS ONE